# Impact of Two Phosphorus Fertilizer Formulations on Wheat Physiology, Rhizosphere, and Rhizoplane Microbiota

**DOI:** 10.3390/ijms24129879

**Published:** 2023-06-08

**Authors:** Kaoutar Bourak, Abdoul Razack Sare, Abdelmounaaim Allaoui, M. Haissam Jijakli, Sébastien Massart

**Affiliations:** 1Terra Research Center, Integrated and Urban Plant Pathology Laboratory, Liege University, Gembloux Agro-Bio-Tech, B-5030 Gembloux, Belgium; kaoutar.bourak@uliege.be (K.B.);; 2Microbiology Laboratory, African Genome Center (AGC), Mohammed VI Polytechnic University, Lot 660, Hay Moulay Rachid, Ben Guerir 43150, Morocco

**Keywords:** phosphorus, microbiota, wheat, rhizo-compartments, growth stage

## Abstract

Phosphorus (P) is the second most important macronutrient for crop growth and a limiting factor in food production. Choosing the right P fertilizer formulation is important for crop production systems because P is not mobile in soils, and placing phosphate fertilizers is a major management decision. In addition, root microorganisms play an important role in helping phosphorus fertilization management by regulating soil properties and fertility through different pathways. Our study evaluated the impact of two phosphorous formulations (polyphosphates and orthophosphates) on physiological traits of wheat related to yield (photosynthetic parameters, biomass, and root morphology) and its associated microbiota. A greenhouse experiment was conducted using agricultural soil deficient in P (1.49%). Phenotyping technologies were used at the tillering, stem elongation, heading, flowering, and grain-filling stages. The evaluation of wheat physiological traits revealed highly significant differences between treated and untreated plants but not between phosphorous fertilizers. High-throughput sequencing technologies were applied to analyse the wheat rhizosphere and rhizoplane microbiota at the tillering and the grain-filling growth stages. The alpha- and beta-diversity analyses of bacterial and fungal microbiota revealed differences between fertilized and non-fertilized wheat, rhizosphere, and rhizoplane, and the tillering and grain-filling growth stages. Our study provides new information on the composition of the wheat microbiota in the rhizosphere and rhizoplane during growth stages (Z39 and Z69) under polyphosphate and orthophosphate fertilization. Hence, a deeper understanding of this interaction could provide better insights into managing microbial communities to promote beneficial plant–microbiome interactions for P uptake.

## 1. Introduction

Phosphorus (P) is the second most essential nutrient for plants’ growth and development. P restriction is an abiotic constraint that severely affects agricultural production and considerably decreases crop productivity [1,2,3]. P is involved in several physiological and metabolic processes, including structural compound formation, energy transfer, cell division and elongation, carbon assimilation, and nitrogen metabolism [4]. P concentrations in soils range from 100 to 3000 mg/kg, whereas the total phyto-available P fraction only ranges from 0.1 to 10 µM, which remains insufficient for plant growth [5,6,7]. However, the widespread use of P fertilizers in agriculture results in the enrichment of heavy metals such as uranium in agricultural ecosystems and the food chain [8], and the eutrophication of groundwater caused by the washout of surplus P can result in toxic algal blooms [9]. To avoid P loss in agricultural fields, it would be preferable to efficiently synchronize P input with crop absorption [10].

Since P fertilization was first patented in 1854, it has been primarily applied in orthophosphate (ortho-P) forms [11]. Unfortunately, ortho-P-based fertilizers have apparent drawbacks regarding the bioavailability of P when they are added to soils, because P in phosphate (PO3–4) form is readily adsorbed by soil clay particles and precipitated by soil cations [12]. Farmers have attempted to cope with this problem by using appropriate P sources: (a) liquid instead of granular phosphate fertilizer [13], (b) applying phosphate fertilizer in split applications [14], and (c) applying phosphate fertilizer with manure [15]. However, none of these approaches can overcome the inherent disadvantages of orthophosphate-based P fertilizers. Furthermore, P is a non-renewable resource; it has been predicted that P consumption will peak in 2030 and that P resources will be depleted within 50–100 years [16,17,18]. Therefore, there is an immediate requirement to improve the P use efficiency (PUE) of all P resources in agriculture, and polyphosphates may be part of this equation. 

Polyphosphate (poly-P) fertilizers are small releases of anionic polymers made up of from two to hundreds of orthophosphate (ortho-P) residues connected by phospho-anhydride bonds to create linear chains (linear poly-P), cyclic structures (meta poly-P), or branching structures (ultra poly-P) [19,20]. Owing to their exceptional biological and chemical degradability [21], and their high solubility [22], increasing attention has been devoted to poly-P fertilizers. Hence, understanding whether P fertilizer formulation affects wheat growth performance, specifically, the morphological and architectural characteristics of the roots, as the root system can maximize nutrient availability [23], is crucial for improving P fertilizer use efficiency. 

Plant growth can be influenced by root-associated microbes which, for example, can play a crucial role in alleviating challenging environmental conditions for plants [24]. Microorganisms living in soils are among the most abundant and varied life forms on earth, playing a part in all global geochemical processes [25]. Plant-associated microorganisms can enhance soil phosphorus availability by increasing root growth and nutrient uptake [26,27], mobilizing organic P through mineralization [28], and triggering microbial nutrient cycling [29,30]. Soil microorganisms can solubilize phosphorus, making it available to plants [31]. This process is known as phosphate solubilization and involves the synthesis of inorganic acids (such as sulphuric, nitric, and carbonic acids) as well as the synthesis of chelating agents [31]. Diverse phosphate-solubilizing microorganisms (PSMs) are crucial in mineralizing organic P, solubilizing inorganic P, and increasing their bioavailability for plant use [32]. The understanding of their biodiversity has been revolutionized through cultivation-independent investigations. The development of high-throughput sequencing technologies and bioinformatics algorithms in recent years has enabled the taxonomic or functional characterization of microbial communities linked to numerous environments [33,34,35]; the most popular technique for microbiome studies is amplicon sequencing for taxonomic characterization [36,37]. 

Bacteria and fungi predominate the soil microbiota, accounting for more than 99% of the microbial biomass in soil samples [38]. Soil bacteria are a source of enzymes and fertilizers because they can reproduce quickly when optimal water, food, and environmental conditions occur, but also, to a lesser extent, under starvation or soil water stress [39]. In addition to bacteria, fungi are the other important components of soil microbiota, playing crucial roles as saprotrophs, plant mutualists, and pathogens [40], and competing with bacteria for access to nutrients through the production of antimicrobial compounds [41]. Furthermore, these bacteria and fungi alter soil phosphorus availability by modulating plant-available phosphorus through mineralization and solubilization [29,32]. They also drive soil nutrient heterogeneity and are critical for nutrient cycling and ecosystem functioning through organic matter decomposition [32,42]. 

The area of soil influenced by plant roots is usually rich in nutrients due to a variety of substances that the roots release into the soil around them, which play a crucial role in promoting plant growth and interactions with soil microorganisms, encouraging root connections with beneficial microbiota [24]. This area can be subdivided into two unique rhizo-compartments: rhizosphere and rhizoplane. The rhizosphere is the part of the soil surrounding plant roots that is actively influenced by plant growth, respiration, and secretions, whereas the rhizoplane is the microbial biofilm attached directly to plant roots [43]. The rhizoplane serves as a regulatory barrier for microbial entry into the host, as there is a compositional change from the bulk soil to the rhizosphere [44].

Rhizospheric microorganisms can modify the plant root system to boost root absorptive capacity and facilitate nutrient uptake when it is limited in the environment [45,46]. Plants adapt to nutrient deficiency by changing root morphology, enlisting the support of microorganisms, and changing the rhizosphere’s chemical environment [47]. Plants depend on beneficial interactions between roots and microbes for nutrient uptake, growth promotion, and disease resistance [48]. Therefore, understanding the relationship between plant roots and rhizospheric microorganisms is crucial for improving plant nutrient uptake.

There is a significant potential to enhance the bioavailability of residual P sources by soil bacteria and fungi. Studies have focused on how phosphorus fertilization changes the diversity and composition of microbial populations in soil and the processes by which these bacteria mine or scavenge soil phosphorus [26,49,50,51,52]. In addition, phosphorus fertilization significantly increases the type and number of genes involved in P cycling [53]. However, it is unclear how P fertilizers’ formulation affects the structure and function of soil microbiota. As a result, a thorough understanding of this interaction may guide how to effectively manage microbial communities to encourage positive plant–microbiome interactions that facilitate P uptake.

The use of an appropriate formulation of phosphate fertilizers (ortho-P vs poly-P) remains a relevant question, the answer to which can have a substantial impact on PUE and therefore on crop yield, to ensure the productivity of sustainable agriculture. Therefore, in this study, we used the two formulations of phosphate fertilizers mentioned above to unravel the effects of each formulation on the physiological parameters of wheat directly related to yield and microbiota, which can be a crucial factor in increasing PUF. For this purpose, wheat was grown in P-deficient soil to measure several physiological traits and an Illumina MiSeq sequencing approach of the bacterial 16S rRNA and fungal ITS1 from different ecological niches and growth stages. The main objective of our study was to use a multidisciplinary approach to study for the first time the effects of two different P fertilizer formulations on wheat physiology and its associated rhizospheric microbiota.

## 2. Results

### 2.1. Effect of P Fertilizer Formulation on Wheat Physiology and Microbiota at the Grain-Filling Stage (Z69)

#### 2.1.1. Wheat Photosynthetic Parameters Performance after Receiving P Fertilizers

We investigated the chlorophyll content throughout wheat growth. We found that poly-P and Ortho-P application positively affected the chlorophyll content index (Figure 1A). At Z69, the relative chlorophyll reached 54.4 ± 11 and 52.5 ± 3 µmol/m^2^ in wheat leaves treated with poly-P and ortho-P, respectively, while it was 26.8 ± 8 µmol/m^2^ in the unfertilized plants (Figure 1A). The maximum rate of relative chlorophyll was measured during the flowering stage, corresponding to Z55, 59.3 ± 5, and 57.6 ± 6 mol/m^2^ in wheat leaves treated with ortho-P and poly-P, respectively. In contrast, it was 23.74 ± 6 mol/m^2^ in unfertilized plants (Figure 1A). Visually, the lack of P in the control was reflected in reduced plant length and symptoms of leaf wilting and yellowish spots on the youngest leaves at Z69 (Appendix A). 

In photosystem I, significant differences were observed under the application of ortho-P and poly-P compared with the control only in the PS1 active centres, starting from Z55 to Z69 (Figure 1B and Appendix A). Nonetheless, no significant differences were detected in the PSII-measured parameters between P fertilizer formulations and the control (Appendix A). No significant differences were observed between the two P formulations.

#### 2.1.2. Shoot Biomass and Nutrient Content

Shoot dry weight was significantly increased by applying both poly-P and ortho-P. The ortho-P application resulted in lower shoot biomass production than the poly-P application, but no significant differences were detected between the two treatments (Table 1). The nutrient content in the leaf organs studied is reported in Table 1, and there were no significant differences between the fertilized plants and the control group.

#### 2.1.3. Root Biomass and Morphological Traits

The present study showed significant effects of P fertilization. At Z69, wheat root trait evaluations under poly-P and ortho-P fertilizers revealed that root morphology was responsive to the P source compared to the control (Figure 2). Analysis of root morphology parameters showed that ortho-P significantly increased root dry weight, length, surface area, and volume by 55%, 101%, 147%, and 210%, respectively, compared to the unfertilized group (Table 2).

#### 2.1.4. Microbiota Diversity Analysis: Dynamics and Composition of Microbial Communities under P Fertilization at Z69

A total of 4,867,000 raw reads were generated from 60 samples (3 treatments × 2 growth stages × 2 rhizo-compartments × 5 replicates) after paired-end sequencing with a length of 300 bp. After filtering, denoising, merging, and chimera removal, high-quality reads were clustered into 10,916 bacterial ASVs and 1251 fungal ASVs (Appendix A). Rarefied feature tables were used to conduct the diversity core metrics (Appendix A), in some cases resulting in less than five replicates by treatment as the sequencing depth for some samples was too low. 

Taxonomic assignment of ASV revealed 36 bacterial phyla in the wheat rhizosphere at the grain-filling stage (Z69). Proteobacteria was the most abundant bacterial phylum (36–38%) in all treatments. Planctomycetota had a maximum of 18% in the poly-P treatment and a minimum of 16% in control. Acidobacteriota phyla accounted for the third highest proportion, ranging between 10% and 12% in the poly-P treatment and the control, respectively (Figure 3A).

Nine fungal phyla were detected in wheat rhizosphere at Z69. As shown in Figure 3B, a large percentage of fungal ASV in the rhizosphere was unidentified in all treatments. A manual BLAST analysis on the NCBI database for a subset of these unidentified sequences showed homologies with uncultured and uncharacterized fungal sequences that are not in UNITE curated database. Ascomycota was the phylum with the highest percentage identified, with a maximum of 38% detected in the ortho-P treatment and a minimum of 18% in the poly-P. This was followed by Mortierellomycota, with a maximum of 12% detected in the control and a minimum of 1.8% detected in the poly-P treatment.

In the rhizoplane, ASV were assigned to 30 bacterial phyla. As in the rhizosphere, Proteobacteria was the most dominant phylum in the rhizoplane in all treatments, with a maximum relative abundance of 48% observed in the ortho-P treatment and a minimum of 43% in control. This was followed by Acidobacteria, with a relative abundance ranging from 10% to 15% detected in the poly-P and the ortho-P treatments, respectively. The third and fourth most abundant phyla were Bacteroidota and Actinobacteria, respectively (Figure 3C).

The taxonomical composition of fungal ASV revealed nine fungal phyla, with the most abundant at the grain-filling stage (Z69) corresponding to Ascomycota, with a maximum average frequency of 47% in the ortho-P treatment. Basidiomycota was the second most dominant phylum (Figure 3D).

Despite the variations observed in the stacked column charts representing taxonomic microbial communities at the phylum level (Figure 3), no taxa showed a significant abundance difference between the treatments using ANCOM analyses.

As for alpha diversity, a pairwise Kruskal–Wallis test revealed that a significant difference was observed for the fungal community in the rhizoplane between the ortho-P treatment and the control at the Z69 (q = 0.04) (Appendix A).

Based on the Bray–Curtis distance metric, visualization of the principal coordinate analysis (PCoA) plot showed clustering of fungal communities under P fertilizer formulations, and the control treatments from the wheat rhizosphere at Z69 (Figure 4A). Statistical analysis based on the PERMANOVA pseudo-F test revealed significant differences between the ortho-P and poly-P (q = 0.04) and also between ortho-P and the control groups (q = 0.04) (Appendix A).

In the rhizoplane visualization of PCoA based on Bray–Curtis distance metric, the clustering of samples showed an effect of P formulations on the fungal community in wheat rhizoplane at Z69 (Figure 4B). In addition, the PERMANOVA pseudo-F test indicated statistically significant differences between fungal populations in the rhizoplane under ortho-P and the control (q = 0.02) (Appendix A). Nevertheless, no significant differences were observed for the individual ASVs.

### 2.2. Effect of P Fertilizer Formulations on Wheat Root-Associated Microbiota at the Tillering Stage (Z39)

In the wheat rhizosphere, the taxonomic assignment of ASV revealed 34 bacterial phyla at the tillering stage (Z39). Proteobacteria was the most abundant bacterial phylum with a maximum relative abundance of 41% detected in the ortho-P treatment at Z39 and a minimum relative abundance of 37% detected in the control. Acidobacteriota was the second most abundant phylum with a maximum relative abundance of 21% in the poly-P treatment and a minimum of 19% in the ortho-P treatment at Z39 (Figure 5A).

Ten fungal phyla were detected in the wheat rhizosphere at Z39, Ascomycota was the phylum with the highest percentage identified with a maximum percentage detected in ortho-P treatment (31%), followed by Mortierellomycota and Chytridiomycota, respectively (Figure 5B). 

ASV were assigned to 32 bacterial phyla in wheat rhizoplane at the tillering stage. Analogous to the bacterial community in the rhizosphere, the most abundant bacterial phylum in the rhizoplane is Proteobacteria, with a maximum relative abundance of 47% detected in the poly-P treatment and a minimum of 41% detected within the control group. This was followed by the abundance of Acidobacteria, with a maximum percentage of 22% detected in the ortho-P treatment. The third and fourth most abundant phyla were the Bacteroidota and the Actinobacteriota, respectively (Figure 5C). 

The taxonomical composition of fungal ASV at the tillering stage (Z39) revealed nine fungal phyla in the rhizoplane, and was dominated by Ascomycota phylum, with a maximum relative abundance of 30% detected in the control group. The second most abundant phylum was Basidiomycota with a maximum relative abundance of 41% in the ortho-P treatment (Figure 5D).

Notwithstanding the variations observed in the stacked column charts, which illustrated the taxonomic microbial communities at the phylum level, the ANCOM analyses failed to detect any taxa that demonstrated statistically significant differences in their abundance between the treatments and the control (Figure 5).

For the PCoA based on Bray–Curtis distance metrics, the clustering of samples showed an effect of P formulations on the bacterial community in wheat rhizosphere at Z39 (Figure 4C). This was confirmed by a pairwise pseudo-F test based on PERMANOVA analyses, revealing that the bacterial community within the ortho-P formulation was significantly different to the bacterial community within poly-P treatment (q = 0.04) and also to that within the control (q = 0.04) (Appendix A). 

### 2.3. Did Wheat Rhizo-Compartment or Growth Stage Shape Microbial Diversity? 

At the grain-filling stage (Z69), the ANCOM test revealed that one bacterial ASV had significant variation in the relative abundances of the wheat rhizo-compartments. An Uncultured Burkholderiales bacterium was more abundant in the rhizosphere than the rhizoplane of wheat at Z69 treated with ortho-P (w = 212). At the same time, ASVs assigned to an uncultured bacterium clone Pyro1 and *Niabella* sp. were more abundant in the rhizosphere than in the rhizoplane of wheat under ortho-P fertilization at Z69 (Appendix A). 

At the tillering stage (Z39), the ANCOM test revealed bacterial ASV with significant variation in the relative abundances of the wheat rhizo-compartments. The uncultured alpha proteobacterium (w = 284) was more abundant in the rhizoplane of wheat treated with poly-P at Z39 than in the rhizosphere. The same test highlighted that an uncultured beta-proteobacterium strain (w = 430) was significantly more abundant in the rhizoplane than in wheat rhizosphere under ortho-P fertilization (Appendix A). 

During the tillering (Z39), significant differences in bacterial alpha and beta diversity were observed between the wheat rhizo-compartments. Based on the Shannon index, a pairwise Kruskal–Wallis test revealed that the bacterial community differed only between the rhizosphere and rhizoplane in wheat under poly-P treatment (q = 0.04) (Appendix A). PERMANOVA pseudo-F statistical analysis for the Bray–Curtis distance metric showed significant differences in the bacterial community between rhizo-compartments in the poly-P group at Z39 (q = 0.01) and the control at Z39 (q = 0.03) (Appendix A). In addition, a clustering was observed between each rhizo-compartment in the bacterial community, indicating differences in the microbial composition (Figure 6A,B).

ANCOM analyses showed that nine bacterial ASV were more abundant in the Z69 growth stage than in the Z39 in wheat rhizosphere treated with poly-P; most of them belonged to the Proteobacteria phylum and were assigned to different genera, including uncharacterized bacteria: *Pseudoxanthomonas* sp. (w = 295), uncultured bacterium clone 2005-MA-5-100207 (w = 294), uncultured alpha proteobacterium (w = 293), bacterium strain GSD10062 (w = 261), *Luteimonas* sp. (w = 249), *Lysobacter* sp. (w = 246), uncultured bacterium clone (w = 144), and uncultured *Lysobacter* sp. (w = 21) (Appendix A). 

Based on the Bray–Curtis distance metric, PERMANOVA pseudo-F test shows that the studied wheat growth stages (Z39 and Z69) had a significant effect on the rhizoplane fungal community in the control group (q = 0.04) (Appendix A). 

Compartment-specific PCoA plots demonstrated growth stage separation in both ecological niches based on the Bray–Curtis distance metric (Figure 7), with significant differences validated by the PERMANOVA pseudo-F test. The differences between tillering and grain filling were more evident in the rhizospheres of all treatments (poly-P: q = 0.02; ortho-P: q = 0.02; and Control: q = 0.01) (Appendix A) than in the rhizoplane (poly-P: q = 0.63; ortho-P: q = 0.02; and Control: q = 0.11) (Appendix A).

## 3. Discussion

The multidisciplinary approach presented in this study allowed us to carry out an in-depth survey of the effects of different P fertilization formulations on wheat holobiont by shedding light on its main physiological traits, and on the dynamics and composition of the rhizosphere and rhizoplane bacterial and fungal communities at different wheat growth stages. 

Furthermore, this study is the first to evaluate the response of bread wheat physiological traits to P fertilization formulations and its root microbiota as a baseline for microbe-enhanced phosphorus plant uptake. 

### 3.1. P Fertilization Stimulated Biomass and Photosynthetic-Based Parameters 

The effectiveness of the photosynthetic process can be studied through chlorophyll fluorescence measurement. Therefore, it could be a valuable instrument to forecast crop P status in reality and assist in determining the correct P treatment to obtain the optimal yield [55,56]. In the present study, applying P, whatever the formulation, enhanced the aboveground characteristics of wheat; this was expected as the soil was initially poor in P. Our results confirmed the role of P in photosynthesis, which has been widely reported [57,58]. More importantly, no significant differences were observed between the formulations of P. Our results support those of [59], where the net rate of chlorophyll content was lower in non-treated wheat than in poly-P and ortho-P. Still, no significant differences were observed between P formulations.

### 3.2. Wheat Roots Are Much More Developed under P Fertilization and Could Be More Efficient in Stressful Situations

Based on our results, we demonstrated that the use of P fertilizers in different formulations significantly affects root morphology and biomass (Table 2 and Figure 2). Our results support earlier research on soil with 17.1 ppm phyto-available P (P-Olsen), which shown that the application of poly-P considerably increased root dry biomass [60]. In addition to playing several other vital roles, roots are highly involved in strengthening the P acquisition efficiency in several crops [61,62]. Our findings partially support earlier research that, in P-deficient soil (6 ppm phyto-available P measured using the Olsen method), the application of various poly-P fertilizers significantly increased both physiological and morphological root traits in comparison to both non-treated durum wheat and ortho-P [63]. This suggests a potential link to the capacity of poly-P to regulate root development for effective phosphorus uptake. Similarly, a growth chamber experiment showed that poly-P application significantly enhanced maize root length compared to ortho-P application [64]. In our study, the poly-P application slightly modified root morphology compared to the ortho-P. However, no significant differences were detected (Table 2). 

In cereal crops, root diameter is a crucial morphological trait that plays a role in many processes, including phosphorus uptake effectiveness [62,65,66,67,68,69]. In our experiment, supplying P fertilizer significantly increased the average diameter of bread wheat compared to the control; however, there were no significant differences between the formulations (Table 2). In contrast, in their study, [63] found significant differences between the ortho-P, the poly-P, and also between the poly-P used fertilizers. 

### 3.3. Influence of P Fertilization Formulations, Wheat Growth Cycle, and Rhizo-Compartment on the Microbiota

Cross-disciplinary studies have recently substantially improved plant–microbiome research, including omics technologies, bioengineering, experimental biology, bioinformatics, and multivariate statistical analysis to generate a quantitative understanding of interactions between the plant and microbiome [70,71,72,73,74].

It is widely known that a variety of biotic and abiotic factors, including compartment niche, plant genetic signal and age, climate, soil type, and nutrients, regulate the assembly and dynamics of microbes in soils and plants [75,76,77,78,79].

In this study, firstly, we reported wheat microbiota’s response to P fertilizer formulations, including polyphosphate studied for the first time, then the rhizo-compartments (rhizosphere and rhizoplane) and the growth stages (Z39 and Z69). Our findings revealed that only one significant difference was observed in the beta diversity represented by the Bray–Curtis index in the bacterial community rhizosphere at Z39 between the ortho-P and control groups (q = 0.042), suggesting a relative stability of the microbiota to the P formulations used. In agreement with our results, a prior investigation of the microbial communities in wheat roots and soil under various agricultural management revealed that the management regime (i.e., conventional and organic managements) had a substantial impact on the bacterial community in the roots but not on the fungal community [80].

Since many agroecosystems are subject to intensive management, agricultural management and fertilization procedures impact the microbial community assembly in the soil–plant continuum [81,82]. Moreover, recent research has shown that crop microbiomes in the rhizosphere, phyllosphere, and endosphere were less susceptible to various fertilization techniques than soil microbiomes [83,84,85]. 

Furthermore, agricultural systems and fertilization approaches significantly influenced microbial co-occurrence patterns in soils and plant compartments [83,84,86].

The duration of fertilizer application also has a strong impact since the crucial function of keystone taxa like Chloroflexi, Nitrospirae, and Mesorhizobiumin in maintaining soil nutrient cycling and crop production after 40 years of fertilization was further supported by empirical evidence from a field long-term fertilization experiment [87]. Additionally, a recent study in a long-term field experiment with different levels of land-use intensity also revealed that the complexity of the microbiome network was significantly influenced by land management, in addition to having an impact on the structure and composition of bacterial, protozoan, and fungal communities [88].

Secondly, we focused on the effect of the rhizo-compartments on microbial diversity. We found that the rhizo-compartments significantly affected the microbiota diversity (Figure 6 and Appendix A). In agreement with our finding, further research revealed that crop-associated microbiomes are mainly sourced from soils and are increasingly enriched and filtered at various plant compartments and during the different plant developmental stages. The rhizosphere and rhizoplane are crucial interfaces for microbial transmission [89]. Growing research on crop plants cultivated in various environments has shown these two plant compartments, or host microhabitats, are the primary determinants of the composition of crop-associated microbiomes [90,91,92,93,94,95,96].

Moreover, Ref. [93] described the microbial communities associated with the soil, rhizosphere, roots, and leaves of sorghum plants and proposed that plant compartment, followed by developmental stage and host genotype, accounted for the majority of variation in fungal communities. In our study, we found that the structure of the bacterial community changed mainly in response to wheat growth stage followed by rhizo-compartments. In contrast, the microbial community composition was stable in response to the P fertilizer formulations used. 

The rhizospheric microbiome of Arabidopsis plants also differs according to their developmental stage (seedling, vegetative, bolting, and flowering) [97] and specific genera like Acidobacteria, Cyanobacteria, Bacteroidetes, Actinobacteria, and specific genera were strongly correlated with the types of root exudates [97]. Their research shows that plants secrete various substances depending on their growth stage, and these exudates help a plant’s microbiome to form specifically for that plant. Similar findings were previously found in the maize soil–plant continuum [89] and the rice root [98], demonstrating that the plant developmental stage significantly impacts microbiome assembly and functioning. Additionally, some Burkholderiaceae, Streptomycetaceae, and Rhizobiaceae bacteria were shown to be highly concentrated at the seedling stage of crop development, and they were recognized as possible helpful microbes of the crop microbiome [89].

Furthermore, a recent study found 26 stable OTUs of Proteobacteria and Actinobacteria that remained throughout the host’s life cycle by examining the root microbiomes of four maize inbred lines from the vegetative stage to the reproductive stage [96]. This study demonstrated that the plant developmental stage significantly impacts root metabolisms and microbiomes by analysing wild-type plants’ root metabolome and ionome at the vegetative and reproductive stages. Indeed, according to research by [99], it was discovered that a series of plant metabolites were deposited throughout the growth of annual grass. This is expected because as plants grow, their physiological needs and exudates’ chemical composition alter [100,101,102,103].

### 3.4. Abundant Taxa and Potential Functional Characteristics for Acquiring Phosphorus

P fertilization has been shown to significantly alter P turnover efficiency by favouring various bacterial groups [104]. In addition, P fertilization can also influence microbial populations and bacterial genes encoding P cycling enzymes involved in P turnover [105]. However, our study found that the microbiota composition remained steady in the face of phosphate fertilizer formulations used.

The present study showed that some taxa were significantly more abundant during the wheat grain-filling stage (Z69) under poly-P fertilization. These taxa were already reported as being PGPR. A study by [106] showed that *Pseudoxanthomonas* sp. could produce siderophores and indoleacetic acid (IAA). In another study by [107], *Pseudoxanthomonas mexicana* was identified as an endophyte from a pear plant and was observed to be positive for IAA production, nitrogen fixation, and solubilization of phosphate. Meanwhile, certain Lysobacter species bacterial strains are a source of biocontrol chemicals capable of protecting plants against illness [108,109].

Our study aimed to better understand bacterial and fungal assembly in the bread wheat rhizo-compartments at different growth stages under different fertilization formulations. We highlighted ASV with significantly different relative abundances between the rhizo-compartments. An ASV from *Burkholderia* sp. was significantly more abundant in the wheat rhizosphere under ortho-P fertilization, this species is known to be essential for symbiotic nitrogen fixation and mycorrhization of plants [110]. As another illustration, *Burkholderia phytofirmans* frequently generate aminocyclopropane-1-carboxylate (ACC) deaminase, which aids in lowering the concentration of the stress hormone ethylene in plants. Therefore, when *B. phytofirmans* is present, the amount of this root elongation inhibitor may be reduced, which may lead to an increase in the development of plant roots [111].

## 4. Materials and Methods

### 4.1. Soil Sampling, Greenhouse Experimentation, and Description of Different P Formulations

The soil utilized in this experiment was collected from a depth of 0-20 cm from a field (50°23′27.5″ N 4°02′39.4″ E) previously sown with maize in the province of Hainaut, southwestern Belgium. The studied soil was characterized as clay-loamy with 1.49 ± 0.04 of P (%), 21.35 ± 1.70 of K (%), 0.23 of total N (%), 4598.998 ± 7.087 of Ca (%), 8.306 ± 0.028 of pH water, 7.826 ± 0.008 of pH KCl, and 12.34 ± 3.71 of organic matter (g/kg). The soil was first air-dried. Subsequently, the stones and small visible plant residues were manually removed, and the remaining soil was ground and passed through an 8 mm sieve before being placed in the pots with sand of 2 mm and containing no P, at a proportion of 3:1 (*v:v*) and a total weight of 2.5 kg. Four bread wheat seeds (Lennox, CRA-W, Gembloux, Belgium) were sown in each pot. 

The experiment was set up in a greenhouse at Gembloux-Agro-Bio-Tech (Liege University). The 16 h photoperiod was applied with an average PPFD of almost 170 μmol m^−2^ s^−1^ daylight in a greenhouse supplemented by an LED light, (Flood Light BX 151) planted, Colasse SA, Seraing, Belgium). Fertilization was performed using poly-P fertilizer with a linear form in a short chain containing 100% poly-P in tripolyphosphate with 47% P2O5. At the same time, the used ortho-P form is a phosphoric acid-based fertilizer containing potassium and 52% of P2O5 with 100% ortho-P. Fertilization doses were calculated based on soil analysis and bread wheat nutrient requirements. Both treatments were fertilized with N, P, and K at 180, 60, and 150 mg/kg of dry soil, respectively. The control group received N and K at the same rate but no P. The experiment followed a completely randomized design with ten replicates (consisting of ten pots containing four plants each) per treatment.

### 4.2. Photosynthetic Parameters 

Photosynthetic parameters were measured during the wheat growth cycle. Five measurements were performed at different wheat growth stages based on Zadoks scale [54]: Z25, Z45, Z55, Z59, and Z69, corresponding to tillering, stem elongation, heading, flowering, and grain filling, respectively. 

The MultispeQ (PhotosynQ Inc, MI, USA) device measured relative chlorophyll content reflecting the amount of chlorophyll in the leaves. The photosystem II (PSII) parameters, including the photosystem II quantum yield (ΦII), which is the portion of light energy that photosystem II absorbs and directs toward photochemistry, where it is used to produce ATP, NADPH, and eventually sugar for plant growth; the non-photochemical exciton quenching (ΦNPQ), which is the amount of light energy absorbed by photosystem II, which is used for non-photochemical quenching and is then lost as heat inside the leaf; and photosystem II photoinhibition (ΦNO), which is typically the amount of light energy lost to uncontrolled processes that might harm the photochemistry. Photosystem I (PSI) parameters, including PSI active centres, PS1 open centres, PS1 over reduced centres, and PS1 oxidized centres. Metadata were then saved on the PhotosynQ platform (https://photosynq.org) before analysis.

### 4.3. Shoot and Root Biomass and Nutrient Content 

Three random samples of the aboveground organs and roots from each treatment were taken from different pots, placed in paper envelopes, and dried at 70 °C for three days to determine the root and shoot dry weights. The nutritional status of the dried powdered shoot samples for P, K, Ca, and Mg were determined by the extraction with NH4-Acetate + EDTA (pH 4.65) and inductively coupled plasma (lCP) according to [112]. The total N content in the shoots was assessed using the standard protocol for dry combustion (NF in ISO 16634-1). 

### 4.4. Root Morphological Traits 

In order to visualize the response of root morphology and architecture to the application of P fertilizer formulations, three entire roots from different pots in each treatment were sampled randomly at Z69, carefully washed, and spread across a plastic box before scanning with an Epson Perfection LA2400 scanner. The data were digitalized by processing scanned root images. The WinRHIZO image analysis system (Regent Instructions, Quebec, QC, Canada) was used to investigate root morphology and related characteristics, focusing on the root length, average root diameter (AvgDiam), volume (RootVolume), and surface area (SurfArea).

### 4.5. Statistical Analysis 

Analysis of variance (ANOVA), followed by Tukey’s honestly significant difference (HSD) test: α = 0.05, was done using SPSS version 20 to examine the effect of two phosphate fertilizer formulations on relative chlorophyll (*n* = 7), photosystem 1 (PS1), and photosystem 2 (PSII) parameters (*n* = 7), root morphological parameters (*n* = 3), shoot and root biomass, and nutrient content (*n* = 3) of bread wheat. 

### 4.6. Microbiota Harvesting and DNA Extraction

In this study, five replicates of each treatment corresponded to pools of two samples. The microbiota were harvested from the rhizosphere and rhizoplane at two wheat growth stages: tillering (Z39) and grain filling (Z69). Briefly, to 2 g of each replicate, 30 mL of KPBT buffer was added and bath sonicated (Ultrasonic cleaner, VWR USC100TH; 45 kHz) for 5 min, as described previously [113]. DNA was extracted using a Fast DNA Spin Kit with Cell Lysis Solution TC (MP Biomedicals, Santa Ana, CA, USA). The manufacturer’s protocol was adapted as recommended by [114]: samples were homogenized using Power-Mix Model L46 (Labinco, Breda, The Netherlands) for 40 s, placed on ice for 2 min, and again homogenized for another 40 s. After that, samples were centrifuged at 14,000× *g* for 10 min to remove cell debris. The DNA extracts were stored at 4 °C until further analysis. Finally, the DNA concentrations were normalized to 15 ng/µL using a Nanodrop ND-1000 spectrophotometer.

### 4.7. Amplicon Library Preparation and Illumina Miseq Sequencing

The Illumina library preparation was performed on the extracted DNA. For the bacterial community, the V3-V4 hypervariable region of the 16S rRNA gene was amplified using the primer pair Bakt_341F/ Bakt_805R [115]. For the fungal community, we used the primer pair ITS1-F_KYO2F/ ITS2_KYO2R [116] to amplify the ITS1 region. Illumina adapters were already linked to both primers.

The final volume of the amplicon PCR was 25 µL, containing 12.5 µL of Kapa HiFi HotStart ReadyMix, 5 µL of 1µL of each primer, and 2.5 µL of DNA normalized using an ND-1000 spectrophotometer nanodrop at 5 ng/µL. For both 16S rRNA and ITS1, PCR was performed at the following temperatures: 95 °C for 5 min, 25 cycles of 95 °C for 30 s, 55 °C for 30 s, and 72 °C for 30 s. The final elongation step was carried out at 72 °C for 5 min. The PCR amplicons were loaded onto a 1.5% agarose gel migrated at 75 V for 45 min and visualized under UV light if the expected bands were obtained. To clean up the PCR products, AMPure XP beads were used according to the manufacturer’s instructions. Illumina tags (indexes) were added using the Nextera XT DNA Library Preparation Kit as described by the manufacturer. The PCR mix contained 5 µL of DNA, 5 µL of each Nextera XT Index Primer, and 25 µL of Kapa HiFi HotStart ReadyMix for a final volume of 50 µL. PCR was run for both 16S rRNA and ITS1 at the following temperatures: 95 °C for 3 min, 8 cycles of (95 °C for 30 s, 55 °C for 30 s, 72 °C for 30 s), and 72 °C for 5 min. A second clean up was performed using AMPure XP beads, according to the manufacturer’s instructions. Both 16S rRNA and ITS1 libraries were further pooled at equal ratio (20 µL:20 µL). The obtained library quantification was performed based on a fluorometric quantification method using a Quant-iT™ PicoGreen™ dsDNA Assay Kit and dsDNA reagents (Invitrogen™). DNA concentration was calculated in nM based on the size of DNA amplicons. The final library was diluted to 4 nM using 10 mM Tris pH 8.5. The libraries were then sent to the GIGA sequencing facilities (Liège University, Liège, Belgium) for paired-end Illumina MiSeq (2 × 300 bp) amplicon sequencing. 

### 4.8. Bioinformatic and Multivariate Statistical Analyses 

Demultiplexing reads and primer trimming were performed at the sequencing centre. The paired-end FASTQ sequences were analysed using QIIME2 version 2020.6.0 [117] (https://qiime2.org). The q2 DADA2 method was used for quality control and feature table construction without trimming sequences [118]. Using the q2 implemented VSEARCH method in the q2 feature-classifier algorithm, ASV (amplicon sequencing variant) were classified. Taxonomical assignments were carried out at 99% of similarity using the reference database SILVA_138 for 16S rRNA reads, and UNIT v7 for fungal reads. The q2 taxa filter-table algorithm was used to discard cytoplasmic contaminations (mitochondria and chloroplast sequences) and to separate bacterial and fungal ASV table. The q2 diversity core-metrics-phylogenetic script was used to generate the alpha- and beta-diversity information. The generated PCoA and boxplots were visualized using the q2 emperor plot tool. 

The q2 diversity alpha-group-significance algorithm was employed for the multivariate statistical analysis based on the Shannon index, and permutational analysis of variance (PERMANOVA 999 permutations) based on Kruskal–Wallis was used. 

The q2 diversity beta-group-significance function was used for the multivariate statistical analysis based on the Bray–Curtis dissimilarity matrix for beta diversity, analyses based on the Bray–Curtis dissimilarities matrix were performed using PERMANOVA (999 permutations) pseudo-F test. 

For all multiple analyse of variances, each PERMANOVA P-value was automatically corrected in QIIME2 with Benjamini–Hochberg correction (FDR). The q2 taxa barplot plug-in was used to generate stacked bar plots of the microbial taxonomic composition. One pseudo count was added to the feature tables before differential abundance tests. The q2 ANCOM plug-in [119] was used to compare differentially abundant features (ASV) among the treatments, growth stage and rhizo-compartments. Q-values with control of the Benjamini–Hochberg correction (FDR) at a 5% type I error rate is already embedded in the ANCOM test before the final significance based on the empirical distribution of a random count variable called w [120]. The raw data obtained from the sequencing process have been stored in the Sequence Read Archive (SRA) at the National Center for Biotechnology Information (NCBI) repository under the accession number PRJNA953618.

## 5. Conclusions

We conducted the first study on the effects of two phosphorous fertilizer formulations on wheat physiology and its associated microbiota. We revealed that poly-P application positively affects wheat physiological traits directly related to yield and significantly influences bacterial and fungal microbiota in the rhizosphere and rhizoplane at different growth stages. 

As a perspective, a deeper understanding of the effect of P formulation fertilizers usually applied on plant-associated microbiota could further serve as a foundation for enhancing PUE, leading to a selection of microorganisms that can increase the PUE. Furthermore, combining bioengineering using rhizospheric microbiome and fertilization technologies using poly-P is an intriguing possibility for improving plant care, an approach that, while still in its infancy, might be of great agricultural relevance. 

## Figures and Tables

**Figure 1 ijms-24-09879-f001:**
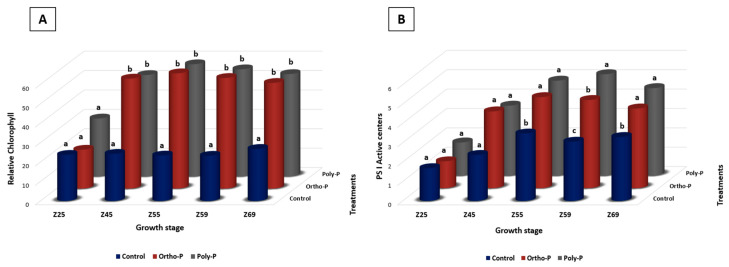
Effect of ortho-P and poly-P application on relative chlorophyll (**A**) and PS1 active centres (**B**) throughout bread wheat growth stages according to [54]. Data are presented as the mean ± SD (*n* = 7). Significant differences between different treatments are indicated by various lowercase letters above the bars. The statistical analysis was determined by a Tukey’s Studentized range (HSD) test: α = 0.05, *n* = 7 using SPSS version 20.0.

**Figure 2 ijms-24-09879-f002:**
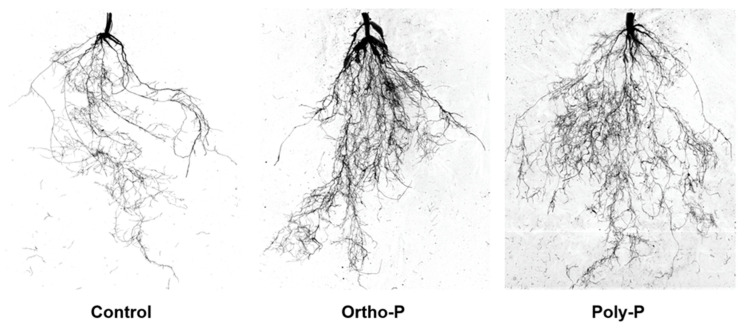
Bread wheat root architecture at Z69 under P application. No P was applied in control.

**Figure 3 ijms-24-09879-f003:**
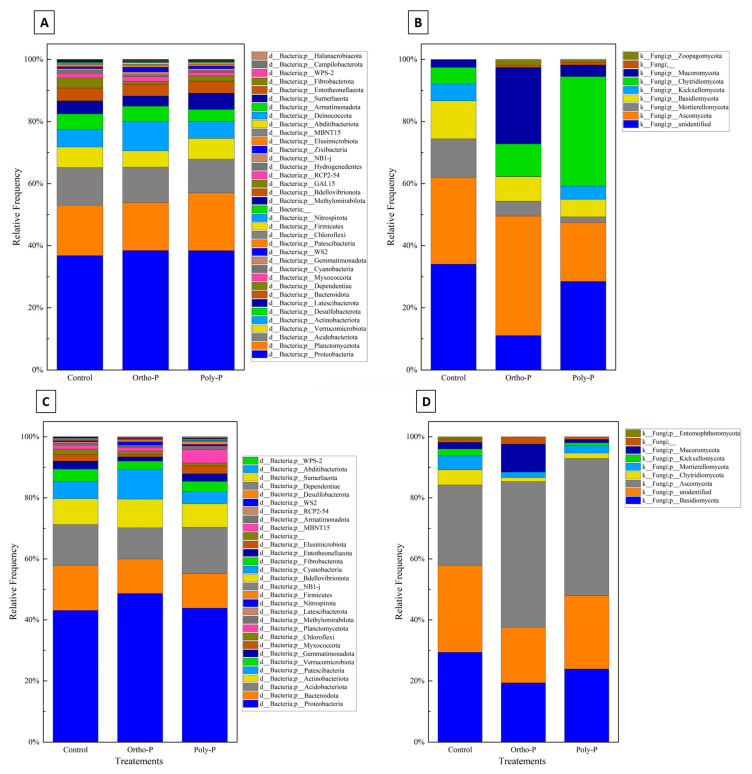
Stacked column charts representing taxonomic at phylum level of the rhizosphere bacterial (**A**), fungal communities (**B**), the rhizoplane bacterial (**C**), and the fungal (**D**) communities of wheat at Z69 in response to P fertilizer formulations.

**Figure 4 ijms-24-09879-f004:**
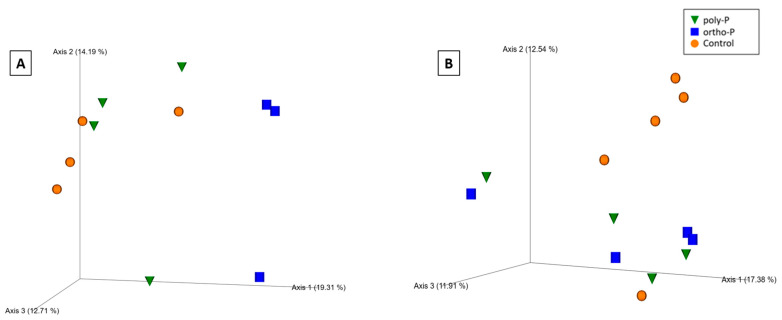
Principal coordinates analysis (PCoA) 3D images based on Bray–Curtis distance of fungi in the rhizosphere (**A**), rhizoplane (**B**) at Z69, and bacteria in the rhizosphere at Z39 (**C**). The clustering observed between each treatment indicates differences in the microbial composition.

**Figure 5 ijms-24-09879-f005:**
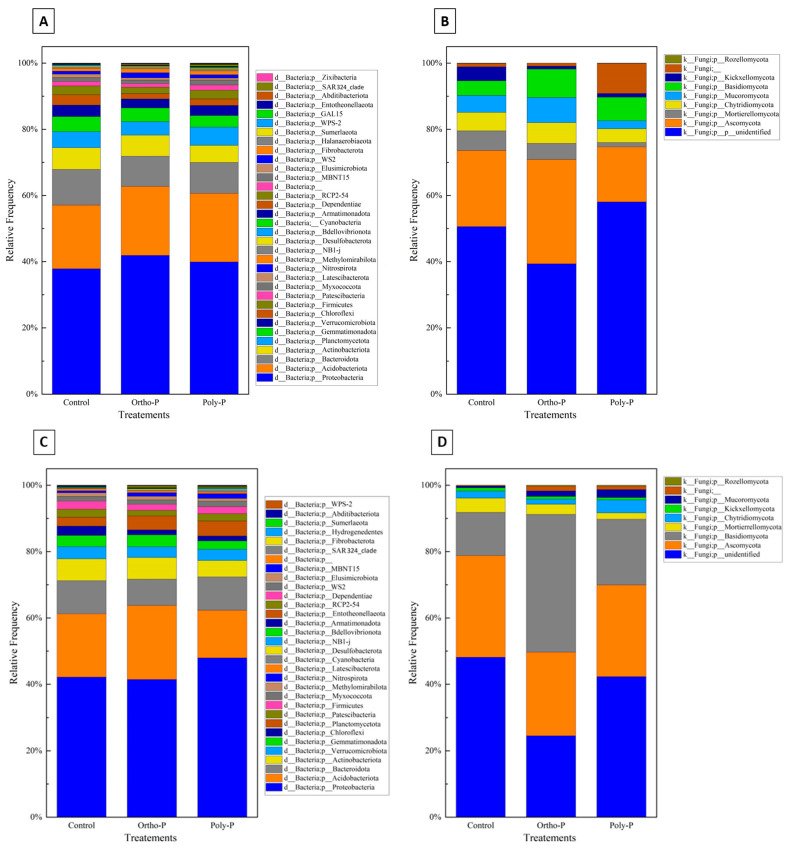
The charts display the different bacterial (**A**,**C**) and fungal (**B**,**D**) phyla found in the rhizosphere and rhizoplane, respectively, of wheat plants at Z39. These charts compare how the bacterial and fungal communities respond to various P fertilizer formulations.

**Figure 6 ijms-24-09879-f006:**
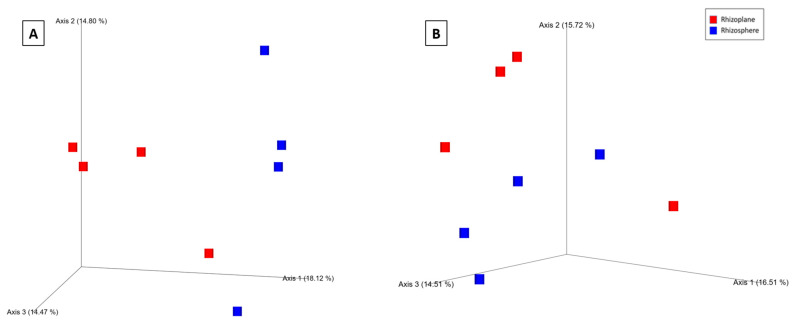
Principal coordinates analysis (PCoA) 3D images based on Bray–Curtis distance of bacteria poly-P at Z39 (**A**) and bacteria in the control at Z39 (**B**). The clustering observed between rhizo-compartments in the bacterial community indicates differences in the microbial composition.

**Figure 7 ijms-24-09879-f007:**
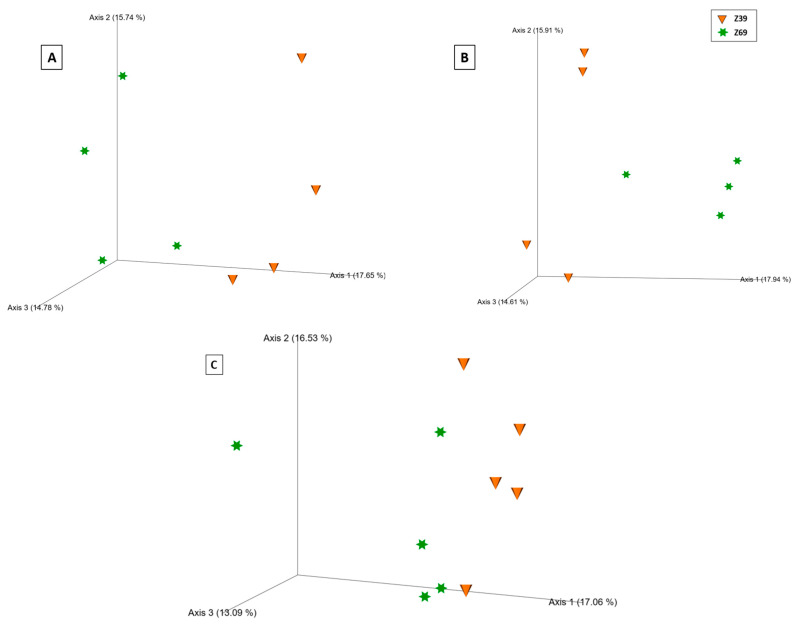
Principal coordinates analysis (PCoA) 3D images based on Bray–Curtis distance of bacteria in the rhizosphere of wheat under poly-P fertilization (**A**), bacteria in the wheat rhizosphere under ortho-P fertilization (**B**), and fungi in the wheat rhizoplane of the control (**C**). The clustering observed between each growth stage in the bacterial and fungal communities indicates differences in microbial composition.

**Table 1 ijms-24-09879-t001:** Shoot dry weight and nutrient content in the leaf organs at the Z69 growth stage determined for the different P fertilizer formulations and the control.

Treatments	Shoot Dry Weight (g)	Nutrients Uptake				
		P (mg/kg)	K (mg/kg)	N (%)	Ca (mg/kg)	Mg (mg/kg)
Control	0.82 ± 0.09 b	2.4 ± 0.04 a	33.5 ± 1.03 a	2.9 ± 0.04 a	8.9 ± 0.08 a	1.4 ± 0.02 a
Ortho-P	1.40 ± 0.08 a	2.4 ± 0.05 a	35.5 ± 0.74 a	2.8 ± 0.07 a	7.8 ± 0.6 a	1.2 ± 0.02 b
Poly-P	1.7 ± 0.10 a	2.506 ± 0.08 a	35.2 ± 0.45 a	2.8 ± 0.06 a	7.6 ± 1.18 a	1.2 ± 0.02 b

Values are means of 3 replicates ± SE; for each column, dissimilar letters imply significant differences at *p* < 0.05, according to Tukey’s test (*p* < 0.05).

**Table 2 ijms-24-09879-t002:** Root biomass and morphological traits of bread wheat under the application of Poly-P and Ortho-P. No P was added to the control group.

Treatments	Root Dry Weight (g)	Root Morphological Traits			
		Length(cm)	SurfArea(cm^2^)	AvgDiam(mm)	RootVolume(cm^3^)
Control	0.29 ± 0.04 b	855 ± 135 b	65 ± 14 b	0.24 ± 0.12 b	0.39 ± 0.12 b
Ortho-P	0.45 ± 0.02 a	1725 ± 210 a	161 ± 18 a	0.30 ± 0.19 a	1.21 ± 0.19 a
Poly-P	0.46 ± 0.04 a	2076 ± 250 a	174 ± 26 a	0.27 ± 0.21 ab	1.17 ± 0.21 a

Means (*n* = 3) that do not share the same letters in the column differ significantly according to Tukey’s test (*p* < 0.05).

## Data Availability

The unprocessed information acquired from the sequencing procedure has been kept in the Sequence Read Archive (SRA) at the National Center for Biotechnology Information (NCBI) repository, identifiable by the accession number PRJNA953618.

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
