# Peer review of "Impact of Two Phosphorus Fertilizer Formulations on Wheat Physiology, Rhizosphere, and Rhizoplane Microbiota"

_ijms, 2023, doi:10.3390/ijms24129879_

Round 1

Reviewer 1 Report

I commend the authors for the analysis methods used, the analysis of the experimental data, the experimental concept and the topic addressed.

I propose the authors to continue and expand the research on other crops and fertilizer formulas (for example: phosphate rock, SSP, TSP) considering the complexity and importance of the field.

Considering the socio-economic importance of the field, the studies should be directed, financed and carried out in the future and validated in experimental field conditions, agricultural funds and different climatic conditions.

Author Response

Dear reviewer, 

Thank you very much for your words and encouragement. We are striving to continue the work already done by exploring additional formulations and combining phosphorus fertilizers with the microbiota to enhance phosphorus utilization efficiency. 

We added a graphic abstract as a reviewer asked, please find the reviewed document attached. 

I remain at your disposal for any further information you may require.

Thank you again for your feedback and guidance.

Sincerely,

Kaoutar BOURAK

Reviewer 2 Report

Far too long article, really difficult to read to the end, of which the authors have not exposed either the limitations or the possible side effects of the study, please to be added.

Also I would appreciate if you would add at least a graphic abstract to simplify the reading and understanding of the text and some concept maps.

Definitely good work, but exposed in a way that is not at all simple and clear, too confusing and complex for a scientific journal.

Author Response

Dear reviewer,

Thank you for your valuable comments. We have tried to take them all into consideration for revision, following your instructions in order to improve our manuscript. Therefore, we have worked on a graphic abstract that can help better understand the study.

In this study, we analyzed the effect of two formulations of phosphorus fertilizers (ortho-P and poly-P) on the wheat holobiome in a P-deficient soil, aiming to select the best formulation for farmers. We initially focused on key physiological traits directly related to yield, including chlorophyll content, photosystems I and II, nutrient content, biomass, and root morphology, using novel phenotyping technologies. We found that regardless of the fertilizer formulation used, the results were significantly better compared to the control, without detecting significant differences between the formulations.

Next, we concentrated on the effect of these formulations on the bacterial and fungal microbiota in the rhizosphere and rhizoplane at different wheat development stages, namely Z39 and Z69 according to the Zadoks scale, using high-throughput sequencing via Illumina Miseq 600, followed by bioinformatic analysis using QIIME2. In this analysis, we focused on taxonomic composition, alpha and beta diversity, aiming to compare between the two fertilizer formulations, development stages, and wheat rhizocompartments. We deduced that the microbiota composition did not change after adding the two formulations, while the microbiota between the studied rhizocompartments and development stages was different. Finally, we performed ANCOM analysis to detect significantly different abundant sequence variants (ASVs) to investigate their role in phosphorus solubilization/hydrolysis for improved phosphorus use efficiency (PUE).

Regarding your question, I hope I’ve understood it correctly, we did not specifically work on the possible side effects of the study since the main objective is to find the best formulation for farmers. We conducted a search to investigate whether there are studies focusing on the side effects of phosphorus fertilizers directly related to physiology and the microbiota but, unfortunately, we did not find any. However, pollution and eutrophication consequences are associated with the use of phosphorus fertilizers. Therefore, we have included a paragraph to demonstrate the harmful effects of phosphorus fertilizers and the need to further enhance their utilization efficiency. If this is what you are asking for, please confirm so that I can permanently incorporate it into the text and provide the references.

Thank you again for your feedback and guidance.

I remain at your disposal for any further information you may require.

Sincerely,

Kaoutar BOURAK

Round 2

Reviewer 2 Report

Well done, it can be accepted.